# SECOND-ORDER REWARDS FOR SUCCESSOR FEATURES

## ABSTRACT

Current Reinforcement Learning algorithms have reached new heights in performance. However, such algorithms often require hundreds of millions of samples, often resulting in policies that are unable to transfer between tasks without full retraining. *Successor features* aim to improve this situation by decomposing the policy into two components: one capturing environmental dynamics and the other modelling reward. Where the reward function is formulated as the linear combination of learned state features and a learned parameter vector. Under this form, transfer between related tasks now only requires training the reward component. In this paper, we propose a novel extension to the *successor feature* framework resulting in a natural second-order variant. After derivation of the new state-action value function, a second additive term emerges, this term predicts reward as a non-linear combination of state features while providing additional benefits. Experimentally, we show that this term explicitly models the environment's stochasticity and can also be used in place of $\epsilon$-greedy exploration methods during transfer. The performance of the proposed extension to the *successor feature* framework is validated empirically on a 2D navigation task, the control of a simulated robotic arm, and the Doom environment.

## 1 INTRODUCTION

Recently, Reinforcement Learning (RL) algorithms have achieved superhuman performance in several challenging domains, such as Atari (Mnih et al., 2015), Go (Silver et al., 2016), and Starcraft II(Vinyals et al., 2019). The main driver of these successes has been the use of deep neural networks, which are a class of powerful non-linear function approximators, with RL algorithms(LeCun et al., 2015). However, this class of Deep Reinforcement Learning (Deep RL) algorithms require immense amounts of data within an environment, often ranging from tens to hundreds of millions of samples(Arulkumaran et al., 2017). Furthermore, commonly used algorithms often have difficulty in transferring a learned policy between related tasks, such as where the environmental dynamics remain constant, but the goal changes. In this case, the model must either be retrained completely or fine-tuned on the new task, in both cases requiring millions of additional samples. If the state dynamics are constant, but the reward structure varies between tasks, it is wasteful to retrain the entire model.

A more pragmatic approach would be to decompose the RL agent's policy such that separate functions can learn the state dynamics and the reward structure; doing so enables reuse of the dynamics model and only requires learning the reward component. *Successor features* (Dayan, 1993) do precisely this; a model-free policy's action-value function is expressed as the dot product between a vector of expected discounted future state occupancies, the successor features, and another vector representing the immediate reward in each of those successor states. The factorization follows from the formulation of the reward as the dot product between a state representation vector and a learned parameter vector, that is a linear product. Therefore, transfer to a new task requires relearning only the reward parameters instead of the entire model and amounts to the supervised learning problem of predicting the current state's immediate reward.

As the reward function of the successor feature framework is linear, it is fair to question whether the model can always accurately predict the reward. As no assumptions are made about the state representation, theoretically it is possible to enable perfect recovery of any reward function if given

predictive state representation Barreto et al. (2017). The state representation, within the successor feature framework, is learned end-to-end by a state encoder to perform well in state reconstruction and reward prediction tasks. The state encoder, given a large enough set of parameters, should have enough representational power to disentangle the factors that are useful for reward prediction by a linear model. However, because of how the encoder is trained, its parameters are utilized for both state reconstruction and reward prediction tasks; while the reward model parameters are only used for reward prediction. If the encoder learns a sub-optimal state representation for reward prediction, say because of a highly complex environment, the reward model might be unable to compensate with its limited set of parameters correctly to predict the reward. Eysenbach et al. (2018) and Hansen et al. (2019) have shown, within the successor feature framework, that there is no strong guarantee that the state encoder is always able to learn features that enable accurate modelling of the reward.

In this paper, a novel extension to the successor feature framework is proposed, where the rewards are modelled with a second-order function. The second-order function, which follows naturally from the original linear variant, gives a stronger guarantee on performance of the model due to both its representational structure and extra parameters. This is especially true in cases where the encoder learns state representations that are less than optimal where a linear model does not have enough representational power to compensate. While, in cases where the encoder can learn sufficient represents for both reconstruction and reward tasks, the second-order function still provides many added benefits. In particular, the additional parameters of the reward model should lessen the representation load of the encoder with regards to the reward tasks allowing more of its representational capacity to be dedicated towards modelling the environment in a task agnostic manner. Further benefits, via a new term emerging after derivation, include a representational form of environmental stochasticity and the ability to use directed exploration during transfer instead of relying on a purely random approach for exploration, such as $\epsilon$-greedy.

Following this, the contributions of this research are as follows:

- A novel formulation of successor features that uses a second-order reward function. This formulation increases the representational power of the reward function while decreasing the representational load on the state encoder providing stronger guarantees on performance.

- Under the new reward formulation, a second term appears that models the future expected auto-correlation matrix of the state features.

- We provide preliminary results that show the second term can be used for guided exploration during transfer instead of relying on $\epsilon$-greedy exploration.

After the introduction of relevant background material in Section 2, we introduce the successor feature framework with a non-linear reward function in Section 3, Section 4 provides experimental support and provides an analysis of the new term in the decomposition. The paper concludes with a final discussion and possible avenues for future work in Section 5.

## 2 BACKGROUND

### 2.1 REINFORCEMENT LEARNING

Consider the interaction between an agent and an environment modelled by a Markov decision process (MDP) (Puterman, 2014). An MDP is defined as a set of states $\mathcal{S}$, a set of actions $\mathcal{A}$, a reward function $R : S \to \mathbb{R}$, a discount factor $\gamma \in [0, 1]$, and a transition function $T : \mathcal{S} \times \mathcal{A} \to [0, 1]$. The transition function gives the next-state distribution upon taking action $a$ in state $s$ and is often referred to as the dynamics of the MDP.

The objective of the agent in RL is to find a policy $\pi$, a mapping from states to actions, which maximizes the expected discounted sum of rewards within the environment. One solution to this problem is to rely on learning a value function, where the *action-value function* of a policy $\pi$ is defined as:

$$Q^\pi(s, a) = \mathbb{E}^\pi \left[ \sum_{t=0}^{\infty} \gamma^t R(s_t) | S_t = s, A_t = a \right]$$

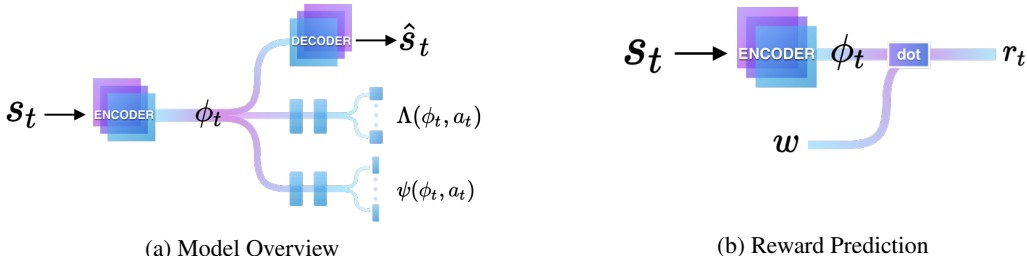

(a) Model Overview                    (b) Reward Prediction

Figure 1: **Model Overview** a) The encoder transforms the raw state into an internal state representation $\phi_t(s)$. The state representation $\phi_t(s)$ is used by the decoder, $\Lambda^\pi(\cdot, a_t)$, and $\psi^\pi(\cdot, a_t)$. The decoder tries to reconstruct the raw input $s_t$ from the state representation $\phi)_t(s)$. $\Lambda^\pi$ and $\psi^\pi$ produce one output per action, with the former predicting matrices and the latter predicting vectors. b) Reward prediction by dotting the current state $\phi_t(s)$, produced by the encoder, and reward weight $w$.

where $\mathbb{E}^\pi[\ldots]$ denotes the expected value when following the policy $\pi$. The policy is learned using an alternating process of *policy evaluation*, given the action-value of a particular policy and *policy improvement*, which derives a new policy that is *greedy* with respect to $Q^\pi(s, a)$(Puterman, 2014).

## 2.2 SUCCESSOR FEATURES

Successor Features (SF) offer a decomposition of the Q-value function and have been mentioned under various names and interpretations(Dayan, 1993; Kulkarni et al., 2016; Barreto et al., 2017; Machado et al., 2017). This decomposition follows from the assumption that the reward function can be approximately represented as a linear combination of learned features $\phi(s; \theta_\phi)$ extracted by a neural network with parameters $\theta_\phi$ and a reward weight vector $w$. As such, the expected one-step reward can be computed as: $r(s, a) = \phi(s, a; \theta_\phi)^\top w$. Following from this, the Q function can be rewritten as:

$$Q(s, a) = \mathbb{E}^\pi \left[ r_{t+1} + \gamma r_{t+2} + \ldots | S_t = s, A_t = a \right]$$

$$= \mathbb{E}^\pi \left[ \phi(a_{t+1}, s_{t+1}; \theta_\phi)^\top w + \phi(a_{t+2}, s_{t+2}; \theta_\phi)^\top w + \ldots | S_t = s, A_t = a \right]$$

$$Q(s, a) = \psi^\pi(s, a)^\top \cdot w$$

where $\psi^\pi(s, a)$ are referred to as the *successor features* under policy $\pi$. The $i^{\text{th}}$ component of $\psi(s, a)$ provides the expected discounted sum of $\phi_t^{(i)}$ when following policy $\pi$ starting from state $s$ and action $a$. It is assumed that the features $\phi(s; \theta_\phi)$ are representative of the state $s$, such that $\psi(.)$ can be turned into a function $\psi^\pi(\phi(s_t; \theta_\phi), a_t)$. For brevity, $\phi(s_t; \theta_\phi)$ is referred to simply as $\phi_t$ and $\psi^\pi(s, a)$ as $\psi(s, a)$.

The decomposition neatly separates the Q-function into two learning problems, for $\psi^\pi$ and $w$: estimating the features under the current policy dynamics and estimating the reward given a state. Because the decomposition still has the same form as the Q-function, the successor features are computed using a Bellman equation update in which the reward function is replaced by $\phi_t$:

$$\psi^\pi(\phi_t, a_t) = \phi_t + \gamma \mathbb{E} \left[ \psi^\pi(\phi_{t+1}, a_{t+1}) \right]$$

such that approximate successor features can be learned using an RL method, such as Q-Learning(Szepesvári, 2009).

Following from this, the approximation of the reward vector $w$ becomes a supervised learning problem. Often, this weight is learned using ordinary least squares from the sampled environmental data. One benefit of having a decoupled representation is that only the relevant function must be relearned when either the dynamics or the reward changes. Therefore, if the task changes, but the environmental dynamics remain constant, only the reward vector parameters $w$ must be relearned, which are minimal compared to the total number of parameters in the full model.

## 3 MODEL, ARCHITECTURE, AND TRAINING

A natural extension to the Successor Feature framework begins by adjusting the fundamental structure of how the reward is represented. In this work, a second-order extension is proposed that improves the flexibility of the reward function while providing other benefits. This paper shows that by improving the representational power of the reward component, with a non-linear function of the state, it provides a stronger guarantee of the framework's performance in such cases by developing a more robust reward component.

This section discusses our change to the successor feature framework, which adjusts the reward function, from a linear function, to a non-linear function. First, a discussion of the new decomposition is given with the full derivation provided in Appendix A. Then experimental support for this change will be presented and analyzed to examine what the new term in the decomposition learns.

### 3.1 NON-LINEAR REWARD FUNCTION

The successor feature framework builds upon functional representation of the current reward $r_t$ as a linear combination of the current state representation $\phi_t(s) \in \mathbb{R}^z$ and a learned reward vector $w \in \mathbb{R}^z$, such that $r_t = \phi_t(s)^\top w$. In this paper we extend the reward model by changing this linear reward model to one with the following form:

$$r_t = \phi_t(s)^\top \mathbf{o} + \phi_t(s)^\top \mathbf{A} \phi_t(s) \tag{1}$$

where $\{\phi_t(s), \mathbf{o}\} \in \mathbb{R}^z$, and $\mathbf{A} \in \mathbb{R}^{z \times z}$. Both $\mathbf{o}$ and $\mathbf{A}$ are learnable parameters modelling the reward structure of the environment. Equation 1 shows that the formulation introduces a non-linear transformation with respect to $\phi_t(s)$. From here on, we use $\phi_t$ instead of $\phi_t(s)$ for brevity. With a slight abuse of notation, we can see the original formulation leads to this if $w$ is replaced with $\mathbf{o} + \mathbf{A}\phi$: $r_t = \phi^\top(\mathbf{o} + \mathbf{A}\phi)$. The state-action value function $Q(s, a)$, under this new reward structure, can be derived to yield:

$$Q^\pi(s_t, a) = \psi^\pi(s_t, a)^\top \mathbf{o} + \beta \mathbf{tr}(\mathbf{A}\Lambda^\pi(s_t, a)) \tag{2}$$

where $\beta \in \{0, 1\}$ controls the inclusion of $\Lambda$ and $\mathbf{tr}$ is the trace operator. It can now be shown that $\psi$ and $\Lambda$ satisfy the Bellman equation(Bellman, 1966):

$$\psi^\pi(s_t, a) = \mathbb{E}^\pi[\phi_{t+1} + \gamma\psi(s_{t+1}, \pi(s_{t+1}))|S_t = s, A_t = a] \tag{3}$$

$$\Lambda^\pi(s_t, a) = \mathbb{E}^\pi[\phi_{t+1}\phi_{t+1}^\top + \gamma\Lambda(s_{t+1}, \pi(s_{t+1}))|S_t = s, A_t = a] \tag{4}$$

where for $\psi$ and $\Lambda$, $\phi$ and $\phi\phi^\top$ respectively play the role of rewards. In addition to $\psi$, it is now necessary to model $\Lambda$, which outputs an $\mathbb{R}^{z \times z}$ matrix *per* action. The quantity $\phi_t\phi_t^\top$ can be interpreted as an auto-correlation matrix of the state features. We can see that this form allows the $\Lambda$ term to model some form of future expected stochasticity of the environment. For example, the diagonal of $\Lambda$ will model a second order moment capturing each feature's change with respect to itself $\phi^1$. We provide analysis and further discussion of $\Lambda$ in Section 4.5.

### 3.2 MODEL STRUCTURE AND TRAINING

The proposed model, shown in Figure 1a, uses an encoder to produce a state embedding $\phi_t$ consumed by downstream modelling tasks. Figure 1b shows how the current reward $r_t$ is predicted using $w$, with $w = \mathbf{o} + \mathbf{A}\phi$, and current state representation $\phi_t$; this process is defined in Equation 1. Similar to previous work with successor features, the structure includes pathways for an encode-decode task and successor feature prediction $\psi$(Machado et al., 2017; Kulkarni et al., 2016; Zhang et al., 2017). The decoder network ensures that the features learned by the encoder, which produces $\phi$, contain useful information for prediction. Furthermore, only the gradients from the state-dependent and reward prediction tasks modify the encoder parameters, and therefore $\phi$. An additional branch is added, by way of the non-linear reward function, to model the quantity $\Lambda(s, a)$. This branch's output is a matrix, which differs from the vector predicting branches $\psi$ and $\phi$.

---

[1]A quantity close to the variance, but not zero mean.

The encode-decode task is trained by minimizing the mean squared difference between the input $s_t$ and the decoder's reconstructed version $\hat{x}_t$ from $\phi$:

$$\mathcal{L}^d(s_t; \theta^\phi, \hat{\theta}^\phi) = [s_t - g(\phi_t; \hat{\theta}^\phi)]^2 \tag{5}$$

where $\phi$ is the output of the encoder with parameters $\theta^\phi$ and $g(\cdot; \hat{\theta}^\phi)$ produces the output of the decoder with parameters $\hat{\theta}^\phi$. As mentioned previously, we train $\psi$ and $\Lambda$, parameterized with $\theta^\psi$ and $\theta^\Lambda$ respectively, using the Bellman equations to minimize the following losses:

$$\mathcal{L}^\psi(s_t, a_t; \theta^\psi) = \mathbb{E}[(\phi_t^- + \gamma\psi(s_{t+1}, a^*; \theta^{-\psi}) - \psi(s_t, a_t; \theta^\psi))^2] \tag{6}$$

$$\mathcal{L}^\Lambda(s_t, a_t; \theta^\Lambda) = \mathbb{E}[(\phi_t^- \phi_t^{-\top} + \gamma\Lambda(s_{t+1}, a^*; \theta^{-\Lambda}) - \Lambda(s_t, a_t; \theta^\Lambda))^2] \tag{7}$$

where $a^* = \max_{a^*} Q(s, a^*)$. To help stabilize learning, we use lagged versions of $\theta^\phi$, $\theta^\psi$, and $\theta^\Lambda$ as done by Mnih et al. (2015); the lagged version is signified with the $-$ symbol in the exponent.

Unfortunately, as the dimensionality of $z$ grows, the number of parameters needed by $\Lambda$ grows quadratically. However, by identifying the $\phi_t \phi_t^\top$ term in $\Lambda(s, a)$ as a symmetric matrix, it is possible to model only the upper triangular portion of the matrix[2], requiring about half the number of parameters. To further reduce parameters, each $\psi$ and $\Lambda$ pathways have two hidden layers before their outputs, reflected in Figure 1a. In this way, the parameters are shared amongst pathways, which contrasts with other works with multiple sets of layers per action $a \in \mathcal{A}$ (Kulkarni et al., 2016; Zhang et al., 2017). To learn the reward parameters $\mathbf{A}$ and $\mathbf{o}$, which are the parameters of the approximated non-linear reward function, the following squared loss function is used:

$$\mathcal{L}^r(s_t; \mathbf{o}, \mathbf{A}) = [r_t - \phi_t^\top \mathbf{o} - \beta\phi_t^\top \mathbf{A}\phi_t]^2 \tag{8}$$

It does not adjust the feature parameters involved in the prediction of $\phi_t$. We factorize matrix $\mathbf{A}$ to reduce its parameters, details are provided in Appendix B. Combining our losses, the composite loss function is the sum of the four losses given above:

$$\mathcal{L}(\theta^\phi, \hat{\theta}^\phi, \theta^\psi, \theta^\Lambda, \mathbf{o}, A) = \mathcal{L}^d + \mathcal{L}^\psi + \beta\mathcal{L}^\Lambda + \mathcal{L}^r \tag{9}$$

In practice, to optimize Equation 9 with respect to its parameters, $(\theta^\psi, \theta^\Lambda)$ and $(\theta^\phi, \hat{\theta}^\phi, o, \mathbf{A})$ are iteratively updated. Doing so increases the stability of the approximations learned by the model and ensures that the branches modelling $\psi$ and $\Lambda$ do not backpropagate gradients to affect $\theta^\phi$ (Machado et al., 2017; Zhang et al., 2017; Kulkarni et al., 2016). Additionally, by training in this way, the state representation $\phi$ can learn features that are both a good predictor of the reward $r_t$ and useful in discriminating between states (Kulkarni et al., 2016).

## 4 EXPERIMENTS

This section examines the properties of the proposed approach on Axes, a navigation task, on Reacher, a robotic control task built using the MuJoCo engine (Todorov et al., 2012), and a 3D maze using the *Doom* game engine. The environments are shown in Figure 2; they each contain tasks specified by goal location and are split between training and test distributions, with the exception of Doom. The environments were chosen as both are similar to tasks in previous work on Successor Features (Barreto et al., 2017) making comparison easier. The Axes and Reacher environments act as a test-bed for our method, allowing us to clearly show that the second-order function provides additional representation capacity to the reward model. To do so, we purposely use a weak encoder, represented by a single hidden layer, that can only learn a sub-optimal state representation.

Within the Axes and Reacher environments, two variants exist, *easy* and *hard*, where each refers to the difficulty of using the state for reward prediction. In both environments the reward function takes the form $-\sqrt{\sum_k^C (g_k - a_k)^2}$ where $g$ is the current goal, $a$ is the agent's perceived location, and $C$ is the co-ordinate system. For Axes $C \in \{x, y\}$ and $C \in \{x, y, z\}$ for Reacher. In the *easy* variant, the state is represented as the distance between the agent and each possible goal within the

---

[2]It would still be necessary to manipulate this matrix so that it forms a full matrix.

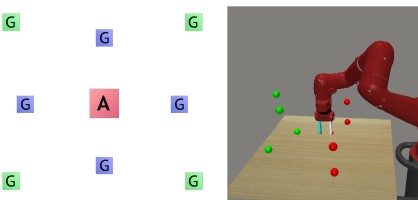 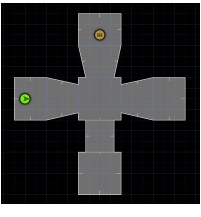 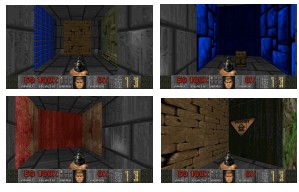

(a) Axes Environment.    (b) Reacher Environment.    (c) Doom Map.    (d) Doom Environment.

Figure 2: **Environments** a) A graphical representation of the Axes environment. The agent, shown as a red square, must traverse to various goal locations marked with the letter "G". The eight goal locations are split between training, shown as blue boxes, and testing, shown as green boxes. b) A rendering of the Reacher task. The agent controls the robotic Sawyer arm to move the end-effector to a 3D point in space. The eight goal locations are shown as balls. Training goals as green, and test goals as red. c) The map layout of the Doom environment. The agent moves between rooms looking for a goal point. d) Images of the Doom environment.

environment, making reward prediction easily accomplished by using a 1-hot reward vector. The *easy* variant is commonly used within other successor feature work Barreto et al. (2017). While, in the *hard* variant, the state is simply the location of the agent and the location of the current goal. Therefore, as the reward involves non-linear functions, a square root and square powers, and the linear variant will have trouble modelling the reward of the *hard* environments with a sub-optimal state representation. The Axes and Reacher environments act as a test-bed to examine the various properties of the second-order function and to show its utility in a controlled environment while examining the properties the second-order extension offers.

To demonstrate the general applicability of our proposed model to complex environments, we evaluate it on a 3D navigation task in the Doom environment from raw pixels[3]. This task is challenging as the model must learn a state representation that is adequate for both reward prediction and use in the successor feature branch from raw pixels.

The primary set of experiments examined the performance between the proposed model and baselines over the training distribution tasks. As transfer to related tasks is a core benefit of the Successor Feature framework, we also evaluate how well the models transfer to unseen tasks from the test distribution on the Axes and Reacher environments. Next, we move to understand the new $\Lambda$ term that appears after derivation of our proposed model. We examine the learned $\Lambda$ function to understand if it captures environmental stochasticity and evaluate different guided exploration strategies using the $\Lambda$ term on new tasks.

## 4.1 ENVIRONMENTS

Additional details of each environment are provided in Appendix C.

**Axes**: In this environment, shown in Figure 2a, the agent, shown by the red square, must traverse the map to reach a goal location using four actions: *up*, *down*, *left*, and *right*. The agent receives a reward equal to the negative distance between itself and the target goal at each step. In the *easy* variant the state $s_t$ is a set of distance between the agent and all available goals, such that $s_t \in \mathbb{R}^8$. While in the *hard* variant, the state $s_t$ is the agent's and the current active goal's 2D coordinates, such that $s_t \in \mathbb{R}^4$.

**Reacher:** The second environment is a control task defined in the MuJoCo physics engine (Todorov et al., 2012), shown in Figure 2b. This environment was chosen to show that the proposed method can scale to difficult control tasks. In this environment, the agent must move a simulated robotic arm to a specific 3D point in space by activating four torque controlled motors. In the *easy* variant the state $s_t$ is a set of distance between the agent and all available goals, such that $s_t \in \mathbb{R}^8$. While in the *hard* variant, the state $s_t$ is the agent's and the current active goal's 2d coordinates, such that $s_t \in \mathbb{R}^4$.

---

[3]Figures used with permission from the author Kulkarni et al. (2016)

| | Axes | | | | Reacher | | | |
|---|---|---|---|---|---|---|---|---|
| | Easy | | Hard | | Easy | | Hard | |
| **Model** | Train | Transfer | Train | Transfer | Train | Transfer | Train | Transfer |
| Random | -3.43 | -3.43 | -3.43 | -3.43 | -78.48 | -78.48 | -78.48 | -78.48 |
| Linear SF | **-1.28** | **-1.29** | -1.58 | -1.6 | **-6.29** | **-6.25** | -10.25 | -10.15 |
| Second-Order SF | -1.3 | -1.45 | **-1.33** | **-1.31** | -6.31 | -6.33 | **-6.87** | **-7.29** |

Figure 3: Performance in the Axes and Reacher environments during training and transfer over the last 1000 steps. Both variants are able to solve the *easy* environment with essentially equal performance. In the *hard* environment the second-order model has higher performance than the linear model, and is much closer to the *easy* variants score.

**Doom:** In this environment, shown in Figure 2 (right), the agent must navigate between 4 rooms trying to collect an item from one of the rooms after which the episode ends. We use the same map as (Kulkarni et al., 2016) but with a slightly different environmental setup, which is detailed in the Appendix. The agent receives a small negative reward per step and a large reward when it picks up the goal item.

## 4.2 EXPERIMENTAL SETUP

The agent is trained on a randomly sampled training tasks; then, during testing, we change to a unseen task. A single policy $\pi$ is trained over all tasks. During transfer to unseen tasks, the model re-learns only the reward parameters, with the remainder of the model frozen (Zhang et al., 2017; Kulkarni et al., 2016). The newly learned policy will vary from the original but is able to exploit previously learned environment knowledge for the new tasks. The training and testing methodology is identical to those used in previous studies (Barreto et al., 2017). A uniform random action baseline was considered in all environments to act as a floor.

Within the Doom environment, we are simply interested in the performance between the second-order model and that of the linear baseline. Therefore, we do not use a train and test split in this environment but instead rely on the randomness of the item and agent spawn locations.

Our method is compared against the successor feature framework with a linear reward model; the architecture is similar to that of Kulkarni et al. (2016). This baseline is identical in all ways to the second-order model except for the exclusion of terms containing $\Lambda$, specifically Equations 2 and 9, which can be obtained by setting $\beta = 0$. More exactly, the linear baseline and our proposed model use the *exact same code* with *only* the $\beta$ and $z$ hyperparameters adjusted.

In all environments we adjust the dimension of the latent representation of $\phi_t$ to ensure that the second-order variant has less than or equal parameters to that of the linear variant. This ensure that the performance from the second-order variant is not from the additional parameters alone. In both the Axes and Reacher environments, the linear variant uses a hidden dimensionality of 24 while the second-order variant has 8. In the Axes, the linear model has $5,704$ parameters and the second-order model has $2,100$ parameters. While, within the Reacher environment this led to $16,710$ and $7,318$ total parameters for the linear and second-order variants, respectively.

In the case of the Doom environment, to again ensure that the performance of the second-order variant is not due to additional parameters, the linear model uses a large three layer convolutional encoder and the second-order model uses a much smaller two layer encoder. This leads to a total of 343M and 276M parameters for the linear and second-order models, respectively. Each model's mean performance is reported on all plots as the average over three runs with varied seeds. Each plot includes the standard deviation over all runs as a shaded area. Additional details can be found in Appendix D.

## 4.3 ENVIRONMENT PERFORMANCE

The primary point of comparison was between the proposed method and the original formulation of the *successor feature* framework, which can be recovered exactly by setting $\beta = 0$ in Equations 2 and 9 of the proposed model. The result of these experiments are shown in Table 3 for both Axes and

Reacher. In both environments, we see that the *easy* task is solved by both the linear and second-order variants. This is expected as the reward, in the case of the linear variant, can be exactly recovered by using a 1-hot encoded reward vector and even with a weaker state representation the linear reward model has enough capacity to compensate. However, we see this is not true in the *hard* task, as the linear variant has weaker performance when compared to the second-order model.

Clearly, the second-order variant provides extra representational capacity to the reward model such that it can compensate on its own for a non-ideal state representation – which is shown by the greater performance on the *hard* tasks. The linear variant, is not able to appropriately model the environments reward structure as the reward is a non-linear function of the state; in this case, the agent's coordinates and the current goal location.

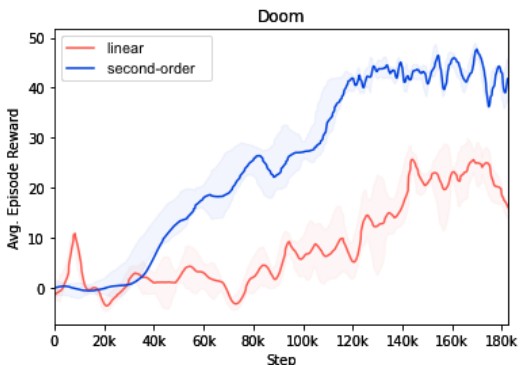

From the result within the Doom environment, shown in Figure 4, we can see that our proposed model is clearly able to out perform the baseline successor feature implementation. Not only does our proposed method is near the ceiling performance of the environment, it converges rapidly. In comparison, the baseline method fails to achieve similar performance and also converges at a much slower rate. From the difference in learning curves we can conclude that

Figure 4: Performance of the baseline linear variant and our proposed second-order model in the Doom environment.

the extra representational power of the reward model has a drastic impact on performance. As both variants have roughly equal parameters, with the linear variant containing more, it can be conclude that the extra representational power is of better use in the reward component of the model instead of the encoder.

## 4.4 TASK TRANSFER

An important property of the *successor feature* framework is the ability to adapt rapidly to new tasks within the same environment. Adaption, or transfer, is accomplished by freezing the model's state-dependent components, such as $\psi$, and quickly learning just the reward parameters $w$. Specifically, we minimize $\mathcal{L}^r(s_t; \mathbf{o}, \mathbf{A}) = [r_t - \phi_t^\top \mathbf{o} + \beta \phi_t^\top \mathbf{A} \phi_t]^2$, of Equation 8, with respect to the parameters $o$ and $\mathbf{A}$ only. Similarly, after training the models to convergence, we change the task distribution within the environment. During transfer, we scaled the learning rate by a fixed factor, randomly initialize the learned weights, and re-decay the $\epsilon$ value so the method has a chance to explore. Full details are provided in Appendix E. The results during transfer on both environments are provided in Table 3.

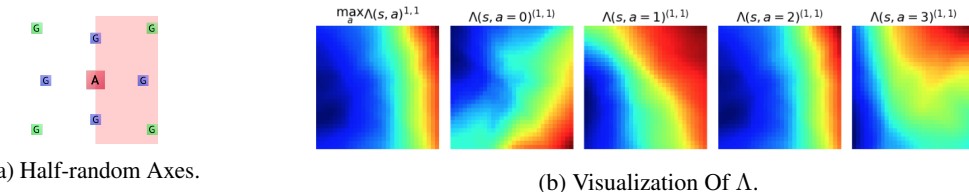

(a) Half-random Axes.

(b) Visualization Of $\Lambda$.

Figure 5: Visualization of Lambda Function on half-random Axes. a) The half-random variant of Axes. b) The learned expected future correlation of one feature with itself along $\Lambda$'s diagonal is visualized over the entire state space. The first column is the max value of $\Lambda$ over the actions. The remaining columns, from left to right, correspond to each action: *left*, *up*, *right*, and *down*. Red and blue correspond to maximal and minimal values.

### 4.5 MODELLING ENVIRONMENTAL STOCHASTICITY

This section examines the $\Lambda$ function to determine whether it can capture stochasticity in the environment in the Axes environment. We use a modified version, referred to as *half-random* and shown in Figure 5a, which is identical in all aspects to the base version except for a location-based conditional that affects the agent's actions. If the agent is within the positive $x$ quadrant of the map, $x > 0$, then actions are randomly perturbed with a fixed probability. Otherwise, they are fully deterministic. After training to convergence on the half-random variant, we examine the $\Lambda$ function that the model has learned. Because the $\Lambda$ function is modelling the auto-correlation matrix, the future expected correlation of each feature with itself is found by looking along the diagonal. Figure 5b shows the result of plotting a diagonal value, in this case, feature $(1, 1)$, of the $\Lambda$-matrix over the entire state space. We can see that one of the diagonal components of $\Lambda$ did indeed learn to approximately model the conditional random field within the environment.

### 4.6 GUIDED EXPLORATION WITH $\Lambda$

Here we examine whether it is possible to use the $\Lambda$-function for guided exploration during transfer within the Axes and Reacher environments.

The Successor Features, given in Equation 2, can be interpreted as predicting the future expected path taken by the policy $\pi$ in an environment. Under this interpretation, $\psi$ can be seen as capturing the expected features of the states and $\Lambda$ the expected variance between state features along these pathways. Adding noise to the $\Lambda$ component would then perturb around the expected path. Therefore, instead of using $\epsilon$-greedy exploration, it is possible to add noise to $\Lambda$ during transfer, such that $\hat{\Lambda}(s, a) = \Lambda(s, a) + \epsilon\Lambda(s, a)$, where $\epsilon$ is sampled from some distribution. During learning, the variance of the sampling distribution, controlled by $\alpha$, can be annealed to some final value. The actions are then sampled from the model at time $t$ as: $a_t = argmax_{a^*}\{\psi(s_t, a^*)^\top o + tr(\mathbf{A}\hat{\Lambda}(s_t, a^*))\}$.

From Figure 6, we see that using $\Lambda$ for guided exploration is indeed a possible alternative to $\epsilon$-greedy. Additionally, we found that using a scalar value sampled from uniform noise, that is $\epsilon \sim \mathcal{U}(-\alpha, \alpha)$ where $\epsilon \in [-\alpha, \alpha]$, provides the best performance.

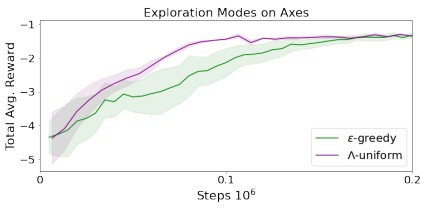

(a) Axes Environment.

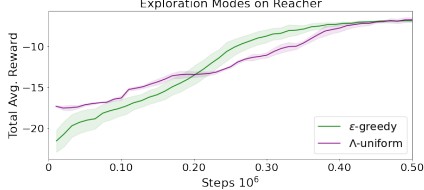

(b) Reacher Environment.

Figure 6: Guided Exploration: The $\Lambda$ component of the proposed model is used to guide exploration during transfer. By using $\Lambda$ the agent explores in directions with large variance in the state space.

## 5 CONCLUSION & FUTURE WORK

In this paper, we have derived a novel formulation of successor features with a non-linear reward. We have shown that the agent can perform well with a second-order reward structure, providing extra flexibility to the reward model. Further, we have shown the utility of the $\Lambda$ term that appears in the derivation of the new state-action function. Experimentally, we have shown that the $\Lambda$ term is able to capture the stochastic nature of an environment and can be used for directed exploration during transfer. In future work, we aim to explore the $\Lambda$ function deeply and if a formulation exists that learns the future expected variance.

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

APPENDIX

## A  NON-LINEAR REWARD DERIVATION

Here we provide the derivation for the non-linear reward in the successor framework. First, we start by assuming the reward $r_t$ has the following form:

$$r_t = \phi_t^\top \mathbf{o} + \phi_t^\top \mathbf{A} \phi_t \tag{10}$$

where $\{\phi_t, \mathbf{o}\} \in \mathbb{R}^{z \times 1}$, and $\mathbf{A} \in \mathbb{R}^{z \times z}$ and both $\mathbf{o}$ and $\mathbf{A}$ are learnable parameters. Following from the definition of the state-action value function $Q(s, a)$, the adjusted reward function can be substituted to yield:

$$Q^\pi(s, a) = \mathbb{E}^\pi[r_{t+1} + \gamma r_{t+2} + \ldots | S_t = s, A_t = a] \tag{11}$$

$$= \mathbb{E}^\pi[\phi_{t+1}^\top \mathbf{o} + \phi_{t+1}^\top \mathbf{A} \phi_{t+1} + \gamma \phi_{t+2}^\top \mathbf{o} + \gamma \phi_{t+2}^\top \mathbf{A} \phi_{t+2} + \ldots | S_t = s, A_t = a] \tag{12}$$

Dropping the conditional portion of the expectation for brevity, linearity of expectation can be used to split apart the terms containing $\mathbf{A}$ and $\mathbf{o}$. Then $\mathbf{o}$ is pulled out from the first term:

$$= \mathbb{E}^\pi[\phi_{t+1}^\top \mathbf{o} + \gamma \phi_{t+2}^\top \mathbf{o} + \ldots] + \mathbb{E}^\pi[\phi_{t+1}^\top \mathbf{A} \phi_{t+1} + \gamma \phi_{t+2}^T \mathbf{A} \phi_{t+2} + \ldots] \tag{13}$$

$$= \mathbb{E}^\pi[\phi_{t+1} + \gamma \phi_{t+2} + \ldots]^\top \mathbf{o} + \mathbb{E}^\pi[\phi_{t+1}^\top \mathbf{A} \phi_{t+1} + \gamma \phi_{t+2}^\top \mathbf{A} \phi_{t+2} + \ldots] \tag{14}$$

By recognizing the first expectation term as the successor features $\psi(s, a)$, Equation 14 can be rewritten as

$$= \psi^\pi(s, a)^\top \mathbf{o} + \mathbb{E}^\pi[\phi_{t+1}^\top \mathbf{A} \phi_{t+1} + \gamma \phi_{t+2}^\top \mathbf{A} \phi_{t+2} + \ldots] \tag{15}$$

Because $\phi^\top A \phi$ results in a scalar, the trace function $tr(\cdot)$ can be used inside the right-hand term:

$$= \psi^\pi(s, a)^\top \mathbf{o} + \mathbb{E}^\pi[\mathbf{tr}(\phi_{t+1}^\top \mathbf{A} \phi_{t+1}) + \mathbf{tr}(\gamma \phi_{t+2}^\top \mathbf{A} \phi_{t+2}) + \ldots] \tag{16}$$

By exploiting the fact that $\mathbf{tr}(\mathbf{AB}) = \mathbf{tr}(\mathbf{BA})$, the terms inside the trace function can be swapped to yield:

$$= \psi^\pi(s, a)^\top \mathbf{o} + \mathbb{E}^\pi[\mathbf{tr}(\mathbf{A} \phi_{t+1} \phi_{t+1}^\top) + \mathbf{tr}(\gamma \mathbf{A} \phi_{t+2} \phi_{t+2}^\top) + \ldots] \tag{17}$$

Because both $tr(\cdot)$ and $\mathbf{A}$ are linear, they can be pulled out of the expectation, giving:

$$= \psi^\pi(s, a)^\top \mathbf{o} + \mathbf{tr}(\mathbb{E}^\pi[\mathbf{A} \phi_{t+1} \phi_{t+1}^\top + \gamma \mathbf{A} \phi_{t+2} \phi_{t+2}^\top + \ldots]) \tag{18}$$

$$= \psi^\pi(s, a)^\top \mathbf{o} + \mathbf{tr}(\mathbf{A} \mathbb{E}^\pi[\phi_{t+1} \phi_{t+1}^\top + \gamma \phi_{t+2} \phi_{t+2}^\top + \ldots]) \tag{19}$$

Finally, the remaining expectation can be expressed as a function:

$$Q^\pi(s, a) = \psi^\pi(s, a)^\top \mathbf{o} + \beta \mathbf{tr}(\mathbf{A} \Lambda^\pi(s, a)) \tag{20}$$

$\beta \in \{0, 1\}$ is a hyperparameter that controls the inclusion of the non-linear component. We define $\psi^\pi$ and $\Lambda^\pi$ as:

$$\psi^\pi(s, a) = \mathbb{E}^\pi[\phi_{t+1} + \gamma \psi(s_{t+1}, \pi(s_{t+1})) | S_t = s, A_t = a] \tag{21}$$

$$\Lambda^\pi(s, a) = \mathbb{E}^\pi[\phi_{t+1} \phi_{t+1}^\top + \gamma \Lambda(s_{t+1}, \pi(s_{t+1})) | S_t = s, A_t = a] \tag{22}$$

## B  $\mathbf{A}$ MATRIX FACTORIZATION

Similar to $\Lambda$, as the dimensionality of $z$ increases, so does the number of parameters needed for modelling matrix $\mathbf{A} \in \mathbb{R}^{z \times z}$. Therefore, in the interest of reducing the number of parameters we use a factorization that splits the matrix $\mathbf{A} \in \mathbb{R}^{z \times z}$ into two parts with a smaller inner dimension $f$, $\mathbf{A} = \mathbf{L} \cdot \mathbf{R}^\top$, where $\{\mathbf{L}, \mathbf{R}\} \in \mathbb{R}^{z \times f}$. By factoring the matrix in this way, we require $2 \times z \times f$ parameters instead of $z \times z$. If we use values for $f$ smaller than $\frac{z}{2}$, we reduce the number of parameters required by matrix $\mathbf{A}$. A similar factorization was suggested in the context of visual question answering (Yu et al., 2017; Fukui et al., 2016). The factorization of $\mathbf{A}$ was primarily done to reduce the total number of parameters in our model.

## C  ENVIRONMENTS

### C.1  AXES

Within this environment eight separate goal locations exist split between train and test distributions. We used a modified version of the robotic model provided by Metaworld (Yu et al., 2019). An episode ends when either the agent reaches the goal or more than 25 steps have elapsed. The agent's starting location is randomly sampled from a grid of $3 \times 3$ step units, centered at $(0, 0)$.

With this state space the agent must learn a reward function that can approximate the distance between itself and the goal location, $d(a, b) = \sqrt{(b_x - a_x)^2 + (b_y - a_y)^2}$, a non-linear function.

### C.2  REACHER

Similarly to Axes, the environment has predefined tasks split between training and test distributions, with the eight goals shown in Figure 2b as green and red balls, respectively. In the Reacher environment, an episode ends when 150 steps have elapsed or the agent is within 7cm of the goal. The agent receives a reward equal to the negative distance between the end-effector and the current target goal at each step.

We discretize the actions such that the agent has nine discrete actions that control the arms movements. Because the models can be used only with discrete actions, it was necessary to transform the environmental actions. Therefore, the four-dimensional continuous action space $\mathcal{A}$ was discretized using two values per dimension: the maximum positive and maximum negative torque for each actuator. An all-zero option was included that applies zero torque along all actuators, resulting in a total of nine discrete actions.

### C.3  DOOM

In the Doom environment the agent must traverse between four rooms looking for a goal. The rooms are separated by doors that the agent must manually open. At each step the agent receives a small negative reward of -0.01 and upon finding the goal it receives +50. The agent perceives the state and the 4 stacked frames of RGB frames of shape (3, 84, 84), corresponding to color channels, width, and height. The agent has 4 actions available: forward, rotate left, rotate right, and activate door. We use an action repeat of 5 across all actions, which differs from the original implementation that used selective action repeat.

## D  EXPERIMENTS

Within both the Axes and Reacher environment, $\epsilon$-greedy was annealed from 1.0 to a final over the first 250k steps.

In the Axes and Reacher environments the encoder and decoder each respectively contain one and two hidden layers with an embedding size equal to the double the raw state size. Initially, on the Axes environment, the models all used the raw features with no encoder such that $\phi_t = \mathbb{I}(s_t)$. We

found this led to worse performance for the linear model as it now had no chance to learn a suitable encoding of the features for reward prediction. Both $\psi$ and $\Lambda$ increase the hidden dimension $z$ by a fixed factor before output. This factor $z_{\text{factor}}$ depends on the environment. All environments used a discount factor of $\gamma = 0.99$, $\lambda = 0.1$, and updated the parameters every $25k$ steps.

### D.1 AXES

An embedding size of $z = 8$ was used for the second-order model and $z = 24$ for the linear model in the Axes environment. The final value used in $\epsilon$-greedy was $0.1$ with a learning rate of $\alpha = 2.5e - 4$ was used. The encoder and decoder in this environment were a single fully connected layer with the same embedding size $z$.

### D.2 REACHER

An embedding size of $z = 8$ was used for the second-order model and $z = 24$ for the linear model in the Reacher environment. The final value used in $\epsilon$-greedy was $0.05$ with a learning rate of $\alpha = 5e - 4$ was used. The encoder and decoder in this environment were a single fully connected layer with the same embedding size $z$.

### D.3 DOOM

An embedding size of $z = 512$ was used for the linear model and $z = 256$ for the second-order model. A encoder network with 3 layers of convolutional layers with hyperparameters (3c, 32o, 8s), (32c, 64o, 4s), and (64c, 64o, 3s) where $c$ is the number of incoming "channels", $o$ is the number of filters, and $s$ is the filter size. The corresponding decoder had 5 layers of transposed convolutional layers: (64c, 256o, 4s), (256c, 128o, 4s), (128c, 64o, 4s), (64c, 32o, 4s), and (32c, 3o, 3s).

While the second-order variant used 2 layer convolutional encoder with: (3c, 16o, 8s), (16c, 32o, 4s). And a corresponding convolutional decoder with 2 layers: (32c, 16o, 4s) and (16c, 3o, 8s).

## E TRANSFER

During transfer we reinitialize the reward specific parameters $\mathbf{A}$ and $\mathbf{o}$ to constant values of $0.1$. All other model parameters are held frozen and do not change. The learning rate is increased by a factor of $2\times$ on the Axes and Reacher environments. We anneal the exploration parameter from 1 to 0.1 in Axes and to 0.05 on the Reacher environments over the first 200k steps.

