# OpenReview forum: "Second-Order Rewards For Successor Features"
_ICLR.cc/2022/Conference — ICLR 2022 Submitted_

### Official Review · Reviewer_gjLL · 2021-10-28

**Correctness:** 3
**Technical Novelty And Significance:** 3
**Empirical Novelty And Significance:** 3
**Recommendation:** 6
**Confidence:** 3

**Main Review:**

The proposed approach requires extra effort and parameters.
Now a matrix needs to be evaluated per action.
Although the authors present favourable results with a smaller number of
parameters, in comparison with the linear model, this seems
misleading. It is relatively easy to increase the number of
parameters in the linear model and still have the same performance.

The authors should include a comparison where the same architecture is
used for both the linear and second order functions, and compare
both approaches in terms of training times, number of parameters, and
performance.

It is not clear why the \beta parameter was introduced and what values
of \beta were used in the experiments.

Another important parameter is the value of zeta. Was it determined by
trial and error? how sensitive is the system to this value?

It is not clear what the authors are trying to prove with Fig. 5,
please clarify.

Typos:
- An extra parenthesis is included in formulation of the Q value
  function in Sec. 2.2:
  \phi(s_{t+1});\theta_{\phi})^Tw => \phi(s_{t+1}_{\phi})^Tw
- In Eq 8, I suppose the minus sign includes both terms: (r - (\phi
... + \beta ...))^2, the same comment applies to Sec. 4.4
- with the only the => with only the
- proposed method near the ceiling => proposed method is near the ceiling


**Summary Of The Paper:**

Successor features decomposes the policy into two components: one
modelling the environmental dynamics and the other, the rewards.
The paper describes an extension to the successor features framework
where the rewards are modelled using a second order function.
The extension provides a more robust reward component.
Under this formulation a second term appears that models the expected
auto-correlation matrix of the state features and can be used for
guided exploration during transfer.
The authors provide experimental results is three domains with
comparison with the linear approach.


**Summary Of The Review:**

The paper describes an extension to the successor feature framework.
Pros:
- a new formulation with provides a more robust reward component
- the approach can be used for exploration during transfer
- a clear improvement over the linear approach with less parameters
Cons:
- it can be seen as an incremental work
- the experiments do not reflect, under the same conditions, the extra
effort needed with the second-order rewards

---

### Official Review · Reviewer_j97Z · 2021-10-28

**Correctness:** 3
**Technical Novelty And Significance:** 3
**Empirical Novelty And Significance:** 2
**Recommendation:** 5
**Confidence:** 4

**Main Review:**

Strengths:
+ the motivation of the paper is clear to me and the proposed method is straightforward and seems to be technically sound.
+ the algorithm was evaluated on a diverse set of environments, with different level of difficulties, including both vectorized state inputs and imagery state inputs.
+ the authors clearly stated and compared the number of parameters used in the proposed algorithm and in the baseline, which is helpful to better gauge the effectiveness of the algorithm. In addition, adjustments were made to reduce the number of total parameters of the proposed neural architecture.

Concerns:

1.[Clarity] The clarity of the paper needs to be further improved in my opinion. Please see the list of items below for details.

+ 1.a. Section 3.1 derives equations (3) and (4) following/conditioning a given policy $\pi$. However, Section 3.2 directly moves to present an algorithm learning $\psi$ and $\Lambda$ in an off-policy Q-learning manner (equations (6) and (7)). It is not clear to me what kind of successor features would be learned with (6) and (7) when the behavior policy keeps changing during training. Could you elaborate a bit more on this? It would be better if the authors could add some explanations or insights in the paper when transitioning from 3.1 (on-policy) to 3.2 (off-policy).

+ 1.b. The analysis of the extra term $\Lambda$ is not sufficient. Related to 1.a, I am not sure how to interpret $\Lambda$ as “the expected variance between state features along these pathways” (described in Section 4.6). Therefore, the exploration strategy in Section 4.6 by randomly perturbing $\Lambda$ is not convincing to me and I am not sure how the exploration is being “guided”. The performance also looks similar to random exploration $\epsilon$-greedy. It could be helpful to compare empirically the trajectories traversed by the agent when using perturbed $\Lambda$ against the trajectories by $\epsilon$-greedy.

+ 1.c. In Section 2.2, first paragraph, it defines reward as a function of state-action pair: $r(s,a) = \phi(s)^T w$, but the RHS and the network architecture Figure 1(b) both do not involve any actions. Please check accordingly in the next version. I believe the paper is modelling reward as a function of state $r(s)$, why not modelling reward as $r(s, a) = \phi(s, a)^T w$ (where a separate feature vector is learned per action)?

2.[Significance + Baselines] Another concern is on the significance of the proposed method when compared with previous approaches in the literature. It would be very beneficial if the authors can add additional baselines to better demonstrate the improvements from the proposed 2nd-order SF framework.

+ 2.a. The paper improves the reward model to better learn and capture the dynamics of the environment via encoder-decoder. Another perspective/direction is improving the “encoder” itself to capture more meaningful state features without modifying the reward model. This may lead to having a more complex encoder module. Existing work [1] showed that adding auxiliary tasks, e.g., predicting the next state, is effective for representation learning, and the architecture in [1] performed well for Atari games. I suggest the authors try adding auxiliary tasks in the linear SF baseline and compare with the 2nd-order SF method. I also encourage the authors to compare against the algorithm proposed in [1] although the focus is on improving exploration with SF. For transfer, I believe [1] used a single FC layer to map features $\phi$ to Q values. It suffices to only retrain the last FC layer when doing task transfer. Hence, the authors could compare transfer as well against [1].

3.[Experiments]

+ 3.a. Why not evaluating the task transfer on the Doom environment? It would be interesting to see transfer results on Doom.

+ 3.b. The visualization of matrix $\Lambda$ in Figure 5 is not very informative to me, because only one element in the matrix (1, 1) is being visualized. In Figure 5(b), does that mean element (1, 1) takes larger values in frequently-visited states? E.g., the region where $x > 0$. It may be helpful if the authors can relate matrix $\Lambda$ or its norm to state visitation counts (previous work [1] has done this before for SF).

+ 3.c. Section 4.6 guided exploration is not clear to me (see 1.b for more comments). The performance looks similar to $\epsilon$-greedy, and it would be interesting to see some results as well on the Doom environment.



Minor Comments:
-	Equation 8: should be $[r_t – A – B]^2$, two minus signs.
-	Section 4.5: feature (1, 1) instead of feature 1?



[1] Machado, Marlos C., Marc G. Bellemare, and Michael Bowling. "Count-based exploration with the successor representation." Proceedings of the AAAI Conference on Artificial Intelligence. Vol. 34. No. 04. 2020.



**Summary Of The Paper:**

The paper provides a new idea for deep reinforcement learning with successor features (SF). The existing SF framework assumes a linear reward model which requires learning predictive and informative state representations to capture reward signals. To reduce the burden of the encoder for learning meaningful state features, this paper extends the linear framework by proposing an additional quadratic term in the reward model, leading to an SF framework with 2nd-order rewards. Since the representation power for modeling rewards is greatly improved, the encoder-decoder can focus more on capturing the dynamics of the environment. Empirically, the 2nd-order SF algorithm outperformed the linear SF baseline, especially when rewards are defined by applying nonlinear transformation of state observations.


**Summary Of The Review:**

The paper presents an extension to deep successor reinforcement learning [2] which adds an extra quadratic term in the reward model, resulting in a new 2nd-order SF framework. The idea is interesting and the motivation is clear. My major concerns are about (1) the clarity of the paper, and (2) the significance of the proposed method, and I suggest the authors include additional baselines and experiments in the rebuttal period (see concerns above for details). Therefore, I don’t think the paper can be accepted with the current version and vote for weak reject.


[2] Kulkarni, Tejas D., et al. "Deep successor reinforcement learning." arXiv preprint arXiv:1606.02396 (2016).

---

### Official Review · Reviewer_yWbZ · 2021-11-03

**Correctness:** 3
**Technical Novelty And Significance:** 2
**Empirical Novelty And Significance:** 2
**Recommendation:** 5
**Confidence:** 4

**Main Review:**

I appreciate the authors clearly wrote the paper and the derivation is relatively clear and appears to be correct.

One major concern I have is that even if we assume reward is a linear function of features, the resulting value function is not hindered and in fact the expressiveness of value function is not limited by the linearity, as features can be an expressive and arbitrarily non-linear function of observations.

Of course, for a fixed trained SF and its value function, there are rewards that are non-linear function of SF, and then the trained model will find it difficult to generalize to such out-of-distribution rewards. But this seems not the focus of this work.

The experiments demonstrate that second-order SF outperforms linear SF on Reacher(Hard) and slighter worse than linear SF on Axes(Easy and Hard) and Reacher(Easy). In hard variants, the states encode less information than easy variants about the agent’s status and goal.
The authors also conducted experiments on pixel based Doom environment and show second-order SF is able to linear SF.

However, I am wondering if feature learning of the linear SF baseline is well tuned and have a fair comparison. The appendix mention that linear SF uses embedding size 512 but second-order SF uses embedding size 256.
There should also have ablation study of the reconstruction auxiliary loss.

The exploration idea is interesting but seems incomplete. The authors wrote the variance term can be seen as “the expected variance between state features along these pathways.” is interesting, but why is adding noise to the variance term helps exploration? To support the claim, the paper would benefit from comparison with related methods on “state features” and “noisy input” e.g. random network distillation, active pretraining with successor features, and noisy network for exploration.

**Summary Of The Paper:**

The authors extend the successor features linear formulation to model a possibly non-linear relationship between features and rewards.

The new method is an instantiation of the successor feature that adds adds features that are dot products of other features, this adds quadratic nonlinearity which the authors argue is helpful to handle more general reward functions.

**Summary Of The Review:**

In summary, the paper needs more clarifications and convincing experiments to support their main claims, especially around why is second order SF is better than linear SF.

---

### Official Review · Reviewer_xH2J · 2021-11-09

**Correctness:** 4
**Technical Novelty And Significance:** 3
**Empirical Novelty And Significance:** 3
**Recommendation:** 6
**Confidence:** 3

**Main Review:**

Strengths
- Prior work / the explanation on how successor feature works is well done.
- The paper is written well and the math is easy to follow. The authors intuitions and assumptions are easy to understand and believable. Replacing the linear model with a quadratic one is intuitive and a natural next step for these kinds of problems.
- The algorithm is evaluated on many different kinds of RL environments which is appreciated and highlights its strengths and weaknesses.

Weaknesses
- The secondary and tertiary contributions of the paper (modelling the auto-correlation matrix of state features and using the second term for guided exploration) seems underdeveloped / incomplete. In Figure 6, for example, neither method significantly outperforms the other and I am not convince that to use their method over epsilon sampling.
- Standard RL model-free baselines would be interesting to see on the experimental setup i.e. where dynamics and reward modelling are not  decoupled. I would expect that model-free methods would outperform the algorithm on standard tasks but would under-perform when it came to transfer learning.


**Summary Of The Paper:**

The authors take the successor feature framework that separately models enivronmental dynamics and reward to use a second order reward learning formulation over the standard linear model. This allows a non-linear relationship to more easily form between features and rewards lessening the burden of a the feature encoder to learn "good" features for the RL task at hand. The authors showcase experiments where the designed new quadratic reward modeler performs better than standard RL systems

**Summary Of The Review:**

Paper is well written and a natural extension to the linear formulation of successor features. Re-implementing the model based on the writing itself should be possible and I thank the authors for clear text.
The latter half of the paper is less well developed coming across as a little undercomplete. I wonder if the paper would benefit more from more significant transfer learning experiments uitlizing the second order model while relegating the auto-correlation matrix / alternative to epsilon exploration to the appendix / future work.

---

### Decision · Program_Chairs · 2022-01-20

**Decision:**

Reject

**Comment:**

The submitted paper considers a form of second-order extension of successor features building on a second-order representation of the reward function in terms of state-features. The authors demonstrate that this approach can be useful for transfer learning and also show an application to exploration.
All reviewers gave borderline recommendations (2x weak accept, 2x weak reject). While most reviewers agree that the proposed approach can be sensible and that the paper is well written, there are concerns that experimental results do not fully support all claims and additional experiments are required to clearly demonstrate advantages over existing baselines. Also the proposed approach for exploration is rather incomplete and not well studied. The raised concerns were not fully refuted by the authors during the discussion period but rather made some reviewers more concerned about full validty of all claims. Thus, while I think the paper has potential and can be turned into a good paper, I am recommending rejection of the paper in its current form. I would like to encourage to authors to carefully address the reviewers' concerns in future versions of the paper.